# Direct Synthesis of Phosphonates and α-Amino-phosphonates from 1,3-Benzoxazines

**DOI:** 10.3390/molecules24020294

**Published:** 2019-01-15

**Authors:** Oscar Salgado-Escobar, Alexis Hernández-Guadarrama, Ivan Romero-Estudillo, Irma Linzaga-Elizalde

**Affiliations:** 1Centro de Investigaciones Químicas-IICBA, Universidad Autónoma del Estado de Morelos, Av. Universidad 1001, 62209 Cuernavaca, Morelos, México; pyrel@hotmail.com (O.S.-E.); alexis13975@hotmail.com (A.H.-G.); 2CONACYT-Centro de Investigaciones Químicas-IICBA, Universidad Autónoma del Estado de Morelos, Av. Universidad 1001, 62209 Cuernavaca, Morelos, México; ivan.romeroest@uaem.mx

**Keywords:** phosphonates, α-aminophosphonates, *o*-quinone methide, *o*-hydroxybenzylic ethers, 1,3-benzoxazines

## Abstract

A straightforward and novel method for transformation of readily available 1,3-benzoxazines to secondary phosphonates and α-aminophosphonates using boron trifluoride etherate as catalyst is developed. The formation of phosphonates proceeds through *ortho*-quinone methide (*o*-QM) generated in situ, followed by a phospha-Michael addition reaction. On the other hand, the α-aminophosphonates were obtained by iminium ion formation and the subsequence nucleophilic substitution of alkylphosphites. This method can be also used for the preparation of *o*-hydroxybenzyl ethers through oxa-Michael addition.

## 1. Introduction

The α-aminophosphonic acids are probably the most important analogues of α-amino acids attributed to their structural analogy obtained by isosteric substitution of planar carboxylic acid (CO_2_H) by tetrahedral phosphonic acid (PO_3_H_2_) [1,2,3]. This kind of compounds have been widely studied and used in agriculture, industry and medicinal chemistry [4,5,6,7,8,9]. In this context, the phosphonates and α-aminophosphonates also constitute an interesting class of compounds which have been utilized in the production of dental additives [10,11], dispersants, corrosion inhibitors [12,13,14], in fire retardants [15,16,17], as well as for preventing deposit formation [18]. Due the different applications, several efforts have been developed for the preparation of phosphonates and α-aminophosphonates [19,20].

In general, the main strategy for the synthesis of phosphonates including the Michaelis-Arbuzov [21] and Michaelis-Becker [22] reactions. On the other hand, the α-aminophosphonates are commonly prepared by Kabachnik-Fields [23,24,25] or Pudovik reactions [26,27]. In this context, Chen [28] and Huang [29] described a practical method for the preparation of *ortho*-hydroxybenzyl phosphonates by phospha-Michael addition of phosphites to *ortho*-quinone methides (*o*-QMs). Particularly, the *o*-hydroxybenzyl phosphonates have been used for the preparation of 1,2-benzoxaphospholes with interesting antioxidants properties [30,31] and as anticancer agents [32].

Considering the high value of these compounds and in connection with our recent work [33], we report herein an innovative methodology for the synthesis of secondary phosphonates and α-aminophosphonates from the reaction of 1,3-benzoxazines with diethyl or triethyl phosphite using catalytic amounts of boron trifluoride etherate. In addition, when the 1,3-benzoxazines was treated with alcohols under reflux conditions provided the corresponding ethers in good yields.

## 2. Results and Discussion

Initially, 1,3-benzoxazines **1a–h** were prepared from the corresponding *2*-(benzylamino)phenols following procedures described in the literature [34,35,36]. In the next step, the study of the reaction conditions for the synthesis of the phosphonate **2b** and α-aminophosphonate **3b** were started. For this purpose, the reaction of the 1,3-benzoxazine **1b** and triethyl phosphite under different conditions (solvents, temperature and using boron trifluoride etherate as catalyst) was examined in order to find the best reaction conditions (Table 1). At first, the 1,3-benzoxazine **1b** was treated with triethyl phosphite in ethanol obtaining the α-aminophosphonate **3b** in 28% yield (Table 1, entry 1). In entry 2 was carried out the reaction at 26 °C in presence of catalytic amounts of boron trifluoride etherate (20 mol%) using DCM as solvent afforded the α-aminophosphonate **3b** in 27% yield. On the other hand, using the same solvent at 40 °C and without catalysts the result was similar (**3b**; 28% yield, entry 3). In the next experiments, using MeCN as solvent at 26 and 82 °C without catalyst, product reaction was not formed (entries 4 and 5).

Alternatively, when MeCN was used in presence of catalytic amounts of boron trifluoride etherate (10 mol%) at 26 °C the phosphonate **2b** in 18% yield was afforded (entry 6). On the other hand, from the reaction of the 1,3-benzoxazine **1b** with triethyl phosphite and increasing amount of boron trifluoride etherate at 20 and 50 mol%, **2b** in 28% yield was obtained in both cases (entries 7 and 8). Then 2.7 equivalents of triethyl phosphite were used and the phosphonate **2b** was isolated in 28% yield (entry 9). In entry 10 the reaction mixture was refluxed in MeCN with the presence of boron trifluoride etherate (20 mol%), from these, the phosphonate **2b** and α-aminophosphonate **3b** in 30 and 47% yield respectively were afforded.

In another experiment, an increase to 3.5 equivalents of the triethyl phosphite under similar conditions did not improve the yield (entry 11). When hexane was used as solvent, no products were observed (Table 1, entry 12).

With these results the formation of the phosphonate was favored when triethyl phosphite, boron trifluoride etherate in catalytic quantities and a polar solvent as acetonitrile at room temperature were used, besides, the reaction was cleaner and the 1,3-benzoxazine that not reacted was recovered.

Under the optimized conditions, the 1,3-benzoxazines **1a**–**h** were reacted with triethyl phosphite in presence of boron trifluoride etherate (20 mol%) in acetonitrile (Scheme 1). When the 1,3-benzoxazines **1b**, **1e** and **1g** were used, the *o*-hydroxybenzyl phosphonates **2b**, **2e** and **2g** were formed in 28–40% yields. The *o*-hydroxybenzyl phosphonates are valuable building block for the synthesis of a wide range of compounds. [29,30,37,38]. From 1,3-benzoxazines **1a**, **1d** and **1h** the α- amino-phosphonates **3a**, **3d** and **3h** were obtained in 6–89% yields (Scheme 1).

The mechanism in Scheme 4 below shows an equilibrium in the ring-opening benzoxazines via iminium ion or *o*-Quinone Methide (*o*-QMs) intermediates. Considering that the stabilization of the iminium ions is directly affected by the steric effect of the substituent (H > Me > *n*-Bu > *s*-Bu > C_6_H_5_ ≥ *p*-ClC_6_H_5_ > *p*-MeC_6_H_4_ > *m*-MeC_6_H_4_), the aminophosphonates with H and Me as substituents were formed in better yields. On the other side, the phosphonates were formed according to the stabilization of the substituent in *o*-QMs intermediates (H> Me> *n*-Bu> *s*-Bu> C_6_H_5_> *p*-ClC_6_H_4_> *p*-MeC_6_H_4_> *m*-MeC_6_H_4_).

In order to study others phosphorus sources, the reaction of the 1,3-benzoxazines **1a**–**h**, diethyl phosphite and boron trifluoride etherate as catalyst in MeCN were carried out (Scheme 2). To our satisfaction only the α-aminophosphonates **3a**–**h** were detected in 24–96% yield. We found that the 1,3-benzoxazines **1a** and **1b** with hydrogen and methyl substituents show the best yields (96 and 80%, respectively), whereas, the 1,3-benzoxazines **1d**, **1g** and **1h** with bulky substituents furnished the α-aminophosphonates **3d**, **3g** and **3h** in moderate yields (Scheme 2). Due to the fact benzyl and *o*-hydroxylbenzyl groups are attached to the nitrogen atom, both move away from each other avoiding the steric hindrance, which causes them to be oriented towards the double bond of the iminium ion inhibiting the access of the phosphite. However, the 1,3-benzoxazine ring opening produces the reaction between the phenolate and hydrogen atom of diethyl phosphite tautomer (Ar-O**^−^**-H-O-P), this facilitate the attack to form the C-P bond, this effect does not occur when triethyl phosphite is used.

With the results obtained in the phosphorylation of *o*-QMs, next we explored the direct transformation of 1,3-benzoxazine **1e**. Thus, **1e** was treated with 3-chloro-1-propanol at 70 °C for 12 h affording the oxa-Michael adduct **5a** in 53 % yield. The ether product is a versatile intermediate to obtain more complex compounds [29,39,40,41] (Scheme 3).

A proposed reaction pathway is depicted in Scheme 4. The formation of α-aminophosphonates can be explained through protonation of the oxygen by the hydrogen of diethyl phosphite which promotes the ring-opening generating the iminium ion, the subsequent phosphorylation provides the corresponding α-aminophosphonates. On the other hand, when triethyl phosphite is used the electronic delocalization of electron pair of nitrogen could generate the ring opening of 1,3-benzoxazines producing the iminium ion (path A) [42,43] which is attacked by the triethyl phosphite to give the α-aminophosphonates. When the oxygen was activated (path B) it promoted *o*-QM formation following by phospha-Michael addition reaction [28] with P(OEt)_3_ to produce the corresponding phosphonates.

## 3. Materials and Methods

### 3.1. General Information

Reagents were obtained from commercial suppliers and were used without further purifification. Melting points were determined in a Fischer Johns apparatus (Pittsburgh, PA, USA) and are uncorrected. NMR spectra were recorded on Varian System instrument (Palo Alto, CA, USA) at 400 MHz for ^1^H- and 100 MHz for ^13^C- and a Varian Gemini at 200 MHz for ^1^H- and 50 MHz for ^13^C-. The spectra were obtained in CDCl_3_ solutions using TMS as an internal reference. ^31^P chemical shifts are reported relative to H_3_PO_4_ as an internal reference. High-resolution CI^+^ and FAB^+^ mass experiments were performed on a JEOL HRMStation JHRMS-700 (Akishima, Tokyo, Japan). The purifification of all compounds was carried out by column chromatography using (silica gel 230–400 mesh). The dichloromethane and acetonitrile were reflfluxed on phosphorous pentoxide and hexane with sodium and benzophenone. Formaldehyde (30%) was used for the reactions.

### 3.2. General Procedure to Obtain the 1,3-benzoxazines 1a-h

A mixture of 2-(benzylamino)-phenol (1.0 eq.) and formaldehyde solution (1.3 eq.) in dichloromethane was stirred at 37 °C for 1 h using a modified Dean-Stark tramp. The crude product was purified by flash chromatography using hexane:EtOAc (99:01) or by recrystallization in methanol.

#### 3.2.1. 3-Benzyl-3,4-dihydro-2H-1,3-benzoxazine (**1a**)

The ^1^H- and ^13^C-NMR data for the compound **1a** were identical to those reported in the literature [36].

#### 3.2.2. 3-Benzyl-4-methyl-3,4-dihydro-2H-1,3-benzoxazine (**1b**)

According to the general procedure, a mixture of 2-{1-(benzylamino)ethyl}phenol (1.0 g, 4.40 mmol) and formaldehyde (0.17 g, 5.72 mmol, 0.46 mL) in dichloromethane (10 mL) was reacted. After purification, **1b** (1.01 g, 99%) was obtained as a colorless oil. ^1^H-NMR (CDCl_3_, 400 MHz): δ 1.46 (d, *J* = 7.2 Hz, 3H), 3.74 (q, *J* = 6.4 Hz, 1H), 3.79 (d, *J* = 13.6 Hz, 1H), 4.01 (d, *J* = 14.0 Hz, 1H), 4.73 (d, *J* = 10.4 Hz, 1H), 5.01 (d, *J* = 10.0 Hz, 1H), 6.83–7.38 (m, 9H). ^13^C-NMR (CDCl_3_, 100 MHz): δ 24.1, 52.8, 56.4, 77.7, 116.8, 120.6, 127.4, 127.7, 128.5, 128.9, 129.0, 129.1, 138.3, 154.3. HRMS (CI^+^): calculated for C_16_H_18_NO [M + H]^+^, *m*/*z* 240.1389; found for [M + H]^+^, *m*/*z* 240.1378.

#### 3.2.3. 3-Benzyl-4-butyl-3,4-dihydro-2H-1,3-benzoxazine (**1c**)

According to the general procedure, a mixture of 2-{1-(benzylamino)pentyl}phenol (1.0 g, 3.71 mmol) and formaldehyde (0.14 g, 4.83 mmol, 0.40 mL) in dichloromethane (15 mL) was reacted. After purification **1c** (1.01 g, 97%) was obtained as a colorless oil. ^1^H-NMR (CDCl_3_, 200 MHz): δ 0.86 (t, *J* = 7.0 Hz, 3H), 1.13–1.88 (m, 6H), 3.48 (dd, *J* = 9.8, 3.8 Hz, 1H), 3.69 (d, *J* = 13.4 Hz, 1H), 4.00 (d, *J* = 13.4 Hz, 1H), 4.68 (dd, *J* = 10.4, 1.6 Hz, 1H), 4.96 (d, *J* = 10.4 Hz, 1H), 6.80–7.35 (m, 9H). ^13^C-NMR (CDCl_3_, 50 MHz): δ 14.2, 22.5, 28.5, 37.9, 56.9, 57.2, 77.8, 116.6, 120.4, 124.9, 127.4, 127.6, 128.4, 128.8, 129.3, 138.7, 153.7. HRMS (CI^+^): calculated for C_19_H_23_NO [M + H]^+^, *m*/*z* 282.1780; found for [M + H]^+^, *m*/*z* 282.1788.

#### 3.2.4. 3-Benzyl-4-(s-butyl)-3,4-dihydro-2H-1,3-benzoxazine (**1d**)

According to the general procedure, a mixture of 2-{1-(benzylamino)-2-methylbutyl}phenol (0.47 g, 1.74 mmol) and formaldehyde (0.06 g, 2.27 mmol, 0.18 mL) in dichloromethane (10 mL) was reacted. After purification **1d** (0.44 g, 91%) was obtained as a colorless oil. ^1^H-NMR (CDCl_3_, 400 MHz): δ 0.83 (t, *J* = 7.2 Hz, 3H), 0.85 (t, *J* = 7.6 Hz, 3H*), 0.94 (d, *J* = 6.4 Hz, 3H), 0.98 (d, *J* = 6.4 Hz, 3H*), 1.15–1.27 (m, 3H), 1.52–1.63 (m, 1H*), 1.69–1.85 (m, 2H*), 3.64 (d, *J* = 13.6 Hz, 1H), 3.66 (d, *J* =13.2 Hz, 1H*), 3.93 (d, *J* = 13.2 Hz, 1H), 3.95 (d, *J* = 13.2 Hz, 1H*), 4.68 (dd, *J* = 10.4, 0.8 Hz, 1H), 4.69 (d, *J* = 10.4 Hz, 1H*), 4.97 (d, *J* = 10.0 Hz, 1H), 5.00 (d, *J* = 10.0 Hz, 1H*), 6.82–7.36 (m, 9H, 9H*). ^13^C-NMR (CDCl_3_, 100 MHz): δ 11.2, 11.9*, 16.1, 16.5*, 25.5, 26.1*, 40.3, 40.8*, 57.5, 57.8*, 61.9, 62.2*, 78.0, 78.5*, 116.5, 116.6*, 116.9, 119.3*, 119.5, 119.7*, 122.4, 122.7*, 127.4, 127.9, 128.0*, 128.4*, 129.4, 129.5*, 130.0, 130.3*, 138.7, 138.8*, 153.7, 154.0*. HRMS (CI^+^): calculated for C_19_H_23_NO [M + H]^+^, *m*/*z* 282.1780; found for [M + H]^+^
*m*/*z* 282.1845.

#### 3.2.5. 3-Benzyl-4-phenyl-3,4-dihydro-2H-1,3-benzoxazine (**1e**)

According to the general procedure, a mixture of 2-{(benzylamino)(phenyl)methyl}phenol (0.80 g, 2.76 mmol) and formaldehyde (0.10 g, 3.58 mmol, 0.30 mL) in dichloromethane (15 mL) was reacted. After crystallization in methanol **1e** (1.04 g, 100%) was isolated as a white solid m.p. = 88–90 °C. ^1^H-NMR (CDCl_3_, 400 MHz): δ 3.91 (d, *J* = 13.6 Hz, 1H), 4.06 (d, *J* = 13.6 Hz, 1H), 4.63 (dd, *J* = 10.4, 1.6 Hz, 1H), 4.76 (d, *J* = 9.6 Hz, 1H), 4.78 (s, 1H), 6.88–7.44 (m, 14H). ^13^C-NMR (CDCl_3_, 100 MHz): δ 56.7, 60.2, 78.1, 116.9, 120.4, 128.3, 128.5, 128.6, 129.2, 129.4, 138.6, 143.6, 154.3. HRMS (CI^+^): calculated for C_21_H_19_NO [M + H]^+^, *m*/*z* 302.1576; found for [M + H]^+^, *m*/*z* 302.1561.

#### 3.2.6. 3-Benzyl-4-(m-tolyl)-3,4-dihydro-2H-1,3-benzoxazine (**1f**)

According to the general procedure, a mixture of 2-{(benzylamino)(*m*-tolyl)methyl}phenol (1.0 g, 3.30 mmol) and formaldehyde (0.12 g, 4.29 mmol, 0.34 mL) in dichloromethane (15 mL) was reacted. After purification **1f** (0.89 g, 86%) was obtained as a colorless oil. ^1^H-NMR (CDCl_3_, 200 MHz): δ 2.31 (s, 3H), 3.93 (d, *J* = 13.4 Hz, 1H), 4.08 (d, *J* = 13.2 Hz, 1H), 4.66 (d, *J* = 10.0 Hz, 1H), 4.77 (s, 1H), 4.82 (d, *J* = 10.0 Hz, 1H), 6.93-7.49 (m, 13H). ^13^C-NMR (CDCl_3_, 50 MHz): δ 21.7, 56.7, 60.4, 78.1, 116.8, 120.4, 126.3, 127.7, 128.4, 128.5, 128.6, 129.5, 129.9, 130.0, 130.2, 130.3, 137.9, 138.5, 143.5, 154.3. HRMS (CI^+^): calculated for C_22_H_21_NO [M + H]^+^, *m*/*z* 317.1780; found for [M + H]^+^, *m*/*z* 317.1830.

#### 3.2.7. 3-Benzyl-4-(p-tolyl)-3,4-dihydro-2H-1,3-benzoxazine (**1g**)

According to the general procedure, a mixture of 2-{(benzylamino)(*p*-tolyl)methyl}phenol (0.3 g, 0.98 mmol) and formaldehyde (0.03 g, 1.27 mmol, 0.10 mL) in dichloromethane (10 mL) was reacted. After crystallization in methanol **1g** (0.23 g, 76%) was obtained as a white solid m.p. = 80–83 °C. ^1^H-NMR (CDCl_3_, 400 MHz): δ 2.31 (s, 3H), 3.91 (d, *J* = 13.2 Hz, 1H), 4.06 (d, *J* = 13.6 Hz, 1H), 4.63 (d, *J* = 10.0 Hz, 1H), 4.78 (s, 1H), 4.82 (d, *J* = 10.0 Hz, 1H), 6.91–7.45 (m, 13H). ^13^C-NMR (CDCl_3_, 50 MHz): δ 21.1, 56.6, 60.4, 78.1, 116.9, 120.4, 126.3, 127.7, 128.5, 128.6, 129.2, 129.5, 129.9, 130.3, 137.2, 138.5, 143.5, 154.4. HRMS (CI^+^): calculated for C_22_H_21_NO [M + H]^+^, *m*/*z* 317.1780; found for [M + H]^+^, *m*/*z* 317.1810.

#### 3.2.8. 3-Benzyl-4-(4-chlorophenyl)-3,4-dihydro-2H-1,3-benzoxazine (**1h**)

According to the general procedure, a mixture of 2-{(benzylamino)(4-chlorophenyl)-methyl}phenol **6** (0.88 g, 2.73 mmol) and formaldehyde (0.09 g, 3.27 mmol, 0.26 mL) in dichloromethane (15 mL) was reacted. After purification **1h** (0.58 g, 64%) was obtained as a colorless oil. ^1^H-NMR (CDCl_3_, 200 MHz): δ 3.88 (d, ^2^*J*_H-H_= 13.2 Hz, 1H), 4.06 (d, ^2^*J*_H-H_= 13.4 Hz, 1H), 4.68 (d, ^2^*J*_H-H_= 11.2 Hz, 1H), 4.69 (dd, ^2^*J*_H-H_= 10.2, 1.6 Hz, 1H), 5.29 (s, 1H), 6.87–7.44 (m, 13H).^13^C-NMR (CDCl_3_, 50 MHz): δ 56.5, 59.3, 77.8, 116.8, 119.5, 120.3, 127.6, 128.2, 128.5, 129.2, 130.0, 130.3, 133.1, 138.2, 141.9, 154.0. HRMS (CI^+^): calculated for C_21_H_18_ClNO [M + H]^+^, *m*/*z* 335.1078; found for [M + H]^+^, *m*/*z* 336.1156.

### 3.3. General Procedure for Preparation of 2-hydroxybenzylphosphonates **2b**, **2e**, **2g** and α-Aminophosphonates **3a**, **3d**, **3h**

A mixture of 1,3-benzoxazine **1a–h** (1.0 eq.), triethyl phosphite (1.0 eq.) and boron trifluoride etherate (20 mol%) in acetonitrile was stirred under nitrogen atmosphere at 26 °C for 72 h. Then, the solvent was evaporated under reduced pressure. The crude was dissolved in dichloromethane (1.0 mL), a saturated solution of ammonium chloride (1.0 mL) was added and the reaction mixture was stirred for 15 min. The organic phase was extracted with dichloromethane and dried with anhydrous sodium sulfate. Finally, the solvent was removed under reduced pressure and the crude was purified by flash chromatography using hexane:EtOAc (80:20).

#### 3.3.1. Diethyl-[1-(2-hydroxyphenyl)ethyl]phosphonate (**2b**)

According to the general procedure, a mixture of 1,3-benzoxazine **1b** (0.10 g, 0.41 mmol), triethyl phosphite 0.06 g (0.41 mmol, 0.07 mL) and boron trifluoride etherate (0.01 g, 0.08 mmol, 0.01 mL) in acetonitrile (3 mL) was reacted. After purification **2b** (0.03 g, 28%) was obtained as a colorless oil. ^1^H-NMR (CDCl_3_, 400 MHz): δ 1.21 (t, *J* = 7.2 Hz, 3H), 1.23 (t, *J* = 7.2 Hz, 3H), 1.57 (dd, *J* = 18.0, 7.6 Hz, 3H), 3.85 (dq, *J* = 23.6, 7.6 Hz, 1H), 3.89-4.08 (m, 4H), 6.87-7.20 (m, 4H). ^13^C-NMR (CDCl_3_, 100 MHz): δ 13.2 (d, ^2^*J*_C-P_ = 4.4 Hz), 16.2, 34.7 (d, ^1^*J*_C-P_ = 136.2 Hz), 63.0 (d, ^2^*J*_C-P_ = 7.3 Hz), 63.1 (d, ^2^*J*_C-P_ = 7.3 Hz), 119.5, 120.9, 124.3 (d, ^2^*J*_C-P_ = 7.4 Hz), 128.7 (d, ^4^*J*_C-P_ = 2.9 Hz), 129.2 (d, ^5^*J*_C-P_ = 7.3 Hz), 155.4 (d, ^3^*J*_C-P_ = 4.4 Hz). ^31^P- NMR (CDCl_3_, 161.90 MHz): δ 31.12. HRMS (CI^+^): calculated for C_12_H_19_O_4_P [M + H]^+^, *m*/*z* 259.1099; found for [M + H]^+^, *m*/*z* 259.1110.

#### 3.3.2. Diethyl-[(2-hydroxyphenyl)(phenyl)methyl]phosphonate (**2e**)

According to the general procedure, a mixture of 1,3-benzoxazine **1e** (0.20 g, 0.66 mmol), triethylphosphite (0.11 g, 0.66 mmol, 0.11 mL) and boron trifluoride etherate (0.01 g, 0.13 mmol, 0.01 mL) in acetonitrile (5 mL) was reacted. After purification **2e** (0.08g, 40%) was obtained as a white solid, m.p. = 157–159 °C. ^1^H-NMR (CDCl_3_, 400 MHz): δ 1.12 (t, *J* = 7.2 Hz, 3H), 1.15 (t, *J* = 7.2 Hz, 3H), 3.74-4.16 (m, 4H), 4.72 (d, *J* = 26.6 Hz, 1H), 6.78-7.53 (m, 9H). ^13^C-NMR (CDCl_3_, 100 MHz): δ 16.3, 18.3, 47.0 (d, ^1^*J*_C-P_ = 136.2 Hz), 63.5 (d, ^2^*J*_C-P_ = 7.0 Hz), 64.0 (d, ^2^*J*_C-P_ = 7.4 Hz), 119.5, 121.0, 123.7, 129.1, 129.5, 129.7, 130.0, 131.1, 131.2, 132.5, 137.2, 155.3. ^31^P-NMR (CDCl_3_, 161.90 MHz): δ 28.78. HRMS (CI^+^): calculated for C_17_H_21_O_4_P [M + H]^+^, *m*/*z* 321.1256; found for [M + H]^+^, *m*/*z* 321.1306.

#### 3.3.3. Diethyl-[(2-hydroxyphenyl)(p-tolyl)methyl]phosphonate (**2g**)

According to the general procedure, a mixture of 1,3-benzoxazine **1g** (0.10 g, 0.31 mmol), triethyl phosphite (0.05 g, 0.31 mmol, 0.05 mL) and boron trifluoride etherate (0.009 g, 0.06 mmol, 0.009 mL) in acetonitrile (3 mL) was reacted. After purification **2g** (0.04 g, 30%) was obtained as a white solid, m.p. = 151–153 °C. ^1^H-NMR (CDCl_3_, 400 MHz): δ 1.13 (t, *J* = 7.2 Hz, 3H), 1.16 (t, *J* = 7.2 Hz, 3H), 2.32 (s, 3H), 3.88 (m, 2H), 4.03 (m, 2H), 4.69 (d, *J* = 26.4 Hz, 2H), 6.79-7.39 (m, 8H). ^13^C-NMR (CDCl_3_, 100 MHz): δ 16.3, 21.2, 47.3 (d, ^1^*J*_C-P_ = 136.2 Hz), 63.4, 64.0, 119.5, 121.0, 123.7, 129.1, 129.5, 129.7, 130.0, 131.1, 131.2, 132.5, 137.2, 155.3. ^31^P-NMR (CDCl_3_, 161.90 MHz): δ 28.78. HRMS (CI^+^): calculated for C_18_H_23_O_4_P [M + H]^+^, *m*/*z* 335.1412; found for [M + H]^+^, *m*/*z* 335.1419.

#### 3.3.4. Diethyl{[benzyl(2-hydroxybenzyl)amino]methyl}phosphonate (**3a**)

According to the general procedure, a mixture of 1,3-benzoxazine **1a** (0.20 g, 0.88 mmol), triethylphosphite (0.16 g, 0.88 mmol, 0.15 mL), and boron trifluoride etherate (0.02 g, 0.17 mmol, 0.02 mL), in acetonitrile (5 mL) was reacted. After purification **3a** (0.28g, 89%) was obtained as a colorless oil ^1^H-NMR (CDCl_3_, 200 MHz): δ 1.29 (t, *J* = 7.0 Hz, 3H), 2.87 (d, *J* = 11.6 Hz, 2H), 3.75 (s, 2H), 3.95, (s, 2H), 4.05 (dq, *J* = 7.2, 7.2 Hz, 4H), 6.72–7.33 (m, 9H). ^13^C-NMR (CDCl_3_, 50 MHz): δ 16.5, 16.6, 47.9 (d, ^1^*J*_C-P_ = 158.3 Hz), 58.8 (d, *J* = 6.4 Hz), 59.1 (d, *J* = 9.9 Hz), 62.2, 62.4, 116.5, 119.5, 121.9, 127.9, 128.7, 129.3, 129.6, 129.9, 136.6, 157.4. ^31^P-NMR (CDCl_3_, 80.9 MHz): δ 21.92. HRMS (CI^+^): calculated for C_19_H_26_NO_4_P [M + H]^+^, *m*/*z* 364.1679; found for [M + H]^+^, *m*/*z* 364.1724.

#### 3.3.5. Diethyl{benzyl[1-(2-hydroxyphenyl)-2-methylbutyl]amino}methyl)phosphonate (**3d**)

According to the general procedure, a mixture of 1,3-benzoxazine **1d,** (0.20 g, 0.83 mmol), triethylphosphite (0.15 g, 0.83 mmol, 0.14 mL) and boron trifluoride etherate (0.02 g, 0.16 mmol, 0.02 mL), in acetonitrile (5 mL) were reacted. After purification **3d** (0.12g, 37%) was obtained as a colorless oil The compound was characterized as diastereomeric mixture. ^1^H-NMR (CDCl_3_, 400 MHz): δ 0.53 (d, *J* = 6.8 Hz, 3H), 0.72 (dd, *J* = 5.2 Hz, 5.2 Hz, 3H), 0.93 (t, *J* = 7.2 Hz, 3H)*, 1.09 (d, *J* = 6.8 Hz, 3H)*, 1.22 (t, *J* = 7.2 Hz, 3H, 3H*), 1.23 (t, *J* = 7.2 Hz, 3H, 3H*), 1.91 (dq, *J* = 7.2, 2.8 Hz, 2H), 1.95 (dq, *J* = 7.2, 2.8 Hz, 2H)*, 2.22–2.35 (m, 1H, 1H*), 3.26 (dd, *J* = 16.8, 3.2 Hz, 1H), 3.28 (dd, *J* = 16.8, 3.2 Hz, 1H)*, 3.32 (d, *J* = 7.6 Hz, 1H, 1H*), 3.59 (d, *J* = 16.0 Hz, 1H), 3.64 (d, *J* = 14 Hz, 2H, 2H*), 3.68 (dd, *J* = 14.0, 4.0 Hz, 1H)*, 4.04–4.14 (m, 2H, 2H*), 4.23–4.33 (m, 2H, 2H*), 6.89–7.33 (m, 9H, 9H*), 10.40 (s, 1H, 1H*). ^13^C-NMR (CDCl_3_, 100 MHz): 10.7, 11.2*, 16.4, 16.5, 16.6, 17.2*, 25.9, 27.3, 33.5, 33.9, 44.9 (d, ^1^*J*_C-P_ = 121.8 Hz), 45.0 (d, ^1^*J*_C-P_ = 122.6 Hz)*, 58.0, 62.1, 62.2*, 71.1, 71.8*, 116.9, 117.1*, 119.0, 119.1*, 122.3, 125.0*, 127.5, 127.5*, 128.5, 128.5*, 128.8, 128.9*, 129.2, 129.2*, 132.9, 138.5*, 148.0, 148.1*. ^31^P-NMR (CDCl_3_, 161.90 MHz): δ 26.01, 26.36* HRMS (CI^+^): calculated for C_23_H_34_NO_4_P [M + H]^+^, *m*/*z* 420.2305; found for [M + H]^+^, *m*/*z* 420.2286.

#### 3.3.6. Diethyl((benzyl((4-chlorophenyl)(2-hydroxyphenyl)methyl)amino)methyl)phosphonate (**3h**)

According to the general procedure, a mixture of 1,3-benzoxazine 1h (0.20 g, 0.59 mmol), triethylphosphite (0.10 g, 0.59 mmol, 0.1 mL) and boron trifluoride etherate (0.01 g, 0.11 mmol, 0.01 mL) in acetonitrile (5 mL) were reacted. After purification 3h (0.01g, 6%) was obtained as a colorless oil ^1^H- NMR (CDCl_3_, 200 MHz): δ 1.29 (t, *J* = 7.2 Hz, 6H), 3.22 (dd, *J* = 16.6, 7.0 Hz, 1H), 3.43 (dd, *J* = 16.5, 4.8 Hz, 1H), 3.73-4.37 (m, 4H), 3.82 (dd, *J* = 6.2, 3.2 Hz, 2H), 4.10 (dd, *J*= 7.9, 7.2 Hz, 1H), 5.06 (s, 1H), 6.69-7.36 (m, 13H), 10.83 (s, 1H). ^13^C-NMR (CDCl_3_, 50 MHz): δ 16.4, 16.5, 45.0 (d, ^1^*J*_C-P_ = 124.5 Hz), 57.6, 61.9 (d, ^3^*J*_C-P_ = 7.5 Hz), 67.0, 122.4, 122.5, 125.6, 127.6, 128.5, 128.7, 129.5, 130.12, 132.5, 133.3, 137.0, 138.0. ^31^P-NMR (CDCl_3_, 161.90 MHz): δ 24.64. HRMS (CI^+^): calculated for C_25_H_29_ClNO_4_P [M + H]^+^, *m*/*z* 474.1603; found for [M + H]^+^, *m*/*z* 474.1653.

### 3.4. General Procedure for Preparation of α-Aminophosphonates **3a**–**h**

A mixture of 1,3-benzoxazines **1a–h** (1.0 eq.), diethyl phosphite (1.0 eq.) and boron trifluoride etherate (0.2 eq.) was stirred at 26 °C for 48 h in acetonitrile, then, the solvent was evaporated under reduced pressure and re-dissolved in dichloromethane. Afterward, a saturated solution of ammonium chloride was added and the reaction mixture was stirred for 15 min. Finally, the organic phase was extracted with dichloromethane and dried over anhydrous sodium sulfate. The solvent was eliminated under reduced pressure and the crude was purified by flash chromatography using hexane:EtOAc (80:20).

#### 3.4.1. Diethyl{[benzyl(2-hydroxybenzyl)amino]methyl}phosphonate (**3a**)

According to the general procedure, a mixture of 1,3-benzoxazine **1a** (0.3 g, 1.33 mmol), diethyl phosphite (0.18 g, 1.33 mmol, 0.17 mL) and boron trifluoride etherate (0.03 g, 0.26 mmol, 0.03 mL) in acetonitrile (5 mL) were reacted during 48 h. The reaction crude was purified by flash chromatography using hexane:EtOAc (80:20). After purification **3a** (0.46 g, 96%) was obtained as a colorless oil.

#### 3.4.2. Diethyl({benzyl[1-(2-hydroxyphenyl)ethyl]amino}methyl)phosphonate (**3b**)

According to the general procedure, a mixture of 1,3-benzoxazine **1b** (0.10 g, 0.41 mmol), diethyl phosphite (0.05 g, 0.41 mmol, 0.07 mL) and boron trifluoride etherate (0.01 g, 0.08 mmol, 0.01 mL) in acetonitrile (3 mL) were reacted. After purification **3b** (0.12 g, 80%) was obtained as a colorless oil. ^1^H- NMR (CDCl_3_, 200 MHz): δ 1.25 (t, *J* = 7.0 Hz, 3H), 1.28 (t, *J* = 7.0 Hz, 3H), 1.47 (d, *J* = 6.6 Hz, 1H), 2.92 (d, *J* = 12.2 Hz, 2H), 3.48 (d, *J* = 12.8 Hz, 1H), 3.79–4.19 (m, 4H), 4.45 (q, *J* = 7.0 Hz, 1H), 6.78–7.34 (m, 9H). ^13^C-NMR (CDCl_3_, 50 MHz): δ 10.3, 16.4, 16.6, 44.2 (d, ^1^*J*_C-P_ = 159.1 Hz), 54.9 (d, ^3^*J*_C-P_ = 7.6Hz), 57.0 (d, ^3^*J*_C-P_ = 6.0 Hz), 62.2 (d, ^2^*J*_C-P_ = 9.1 Hz), 62.4 (d, ^2^*J*_C-P_ = 7.5 Hz), 116.7, 119.3, 126.5, 127.3, 127.8, 128.6, 129.0, 129.8, 137.0, 157.1. ^31^P-NMR (CDCl_3_, 80.9 MHz): δ 26.19. HRMS (CI^+^): calculated for C_20_H_28_NO_4_P [M + H]^+^, *m*/*z* 378.1756; found for [M + H]^+^, *m*/*z* 378.1783.

#### 3.4.3. Diethyl({benzyl[1-(2-hydroxyphenyl)pentyl]amino}methyl)phosphonate (**3c**)

According to the general procedure, a mixture of 1,3-benzoxazine **1c** (0.30 g, 1.06 mmol), diethyl phosphite (0.14 g, 1.06 mmol, 0.13 mL) and boron trifluoride etherate (0.03 g, 0.21 mmol, 0.03 mL) in acetonitrile (5 mL) were reacted. After purification **3c** (0.26 g, 58%) was obtained as a colorless oil. ^1^H -NMR (CDCl_3_, 200 MHz): δ 0.90 (t, *J* = 7.2 Hz, 3H), 1.27 (t, *J* = 6.8 Hz, 3H), 1.28 (t, *J* = 7.2 Hz, 3H), 1.37 (q, *J* = 7.2 Hz, 2H), 1.78-1.88 (m, 2H), 1.95–2.05 (m, 2H), 2.91 (d, *J* = 16.0 Hz, 1H), 3.00 (dd, *J* = 16.0, 7.2 Hz, 1H), 3.64 (d, *J* = 13.2 Hz, 1H), 3.92–4.07 (m, 4H), 4.03 (d, *J* = 13.2 Hz, 1H), 4.15 (d, *J* = 10.0 Hz, 2H), 6.81-7.34 (m, 9H), 9.60 (s, 1H). ^13^C-NMR (CDCl_3_, 100 MHz): δ 14.2, 16.5, 16.6, 23.1, 26.0, 29.5, 44.4 (d, ^1^*J*_C-P_ = 155.3 Hz), 55.1 (d, ^3^*J*_C-P_ = 7.3 Hz), 62.1 (d, ^2^*J*_C-P_ = 5.8 Hz), 62.3 (d, ^2^*J*_C-P_ = 5.9 Hz), 63.2, 117.2, 119.2, 125.2, 127.8, 128.5, 128.7, 128.9, 129.9, 137.6, 157.3. ^31^P-NMR (CDCl_3_, 161.90 MHz): δ 25.71. HRMS (CI^+^): calculated for C_23_H_34_NO_4_P [M + H]^+^, *m*/*z* 420.2305; found for [M + H]^+^, *m*/*z* 420.2355.

#### 3.4.4. Diethyl{benzyl[1-(2-hydroxyphenyl)-2-methylbutyl]amino}methyl)phosphonate (**3d**)

According to the general procedure, a mixture of 1,3-benzoxazine 1d (0.30 g, 1.06 mmol), diethyl phosphite (0.14 g, 1.06 mmol, 0.13 mL) and boron trifluoride etherate (0.03 g, 0.21 mmol, 0.03 mL) in acetonitrile (5 mL) was reacted. After purification 3d (0.16 g, 37%) was obtained as a colorless oil. The compound was characterized as diastereomeric mixture.

#### 3.4.5. Diethyl({benzyl[(2-hydroxyphenyl)(phenyl)methyl]amino}methyl)phosphonate (**3e**)

According to the general procedure, a mixture of 1,3-benzoxazine **1e** (0.20 g, 0.66 mmol), diethyl phosphite (0.09 g, 0.66 mmol, 0.08 mL) and boron trifluoride etherate (0.018 g, 0.13 mmol, 0.016 mL) in acetonitrile (5 mL) was reacted. After purification **3e** (0.19 g, 65%) was obtained as s colorless oil. ^1^H-NMR (CDCl_3_, 400 MHz): δ 1.25 (t, *J* = 7.2 Hz, 3H), 1.31 (t, *J* = 7.2 Hz, 3H), 2.86 (dd, *J* = 15.6, 5.6 Hz, 1H), 2.99 (dd, *J* = 18.0, 15.6 Hz, 1H), 3.64 (d, *J* = 13.2 Hz, 1H), 4.14 (d, *J* = 12.8 Hz, 1H), 3.85–4.00 (m, 2H), 4.04–4.14 (m, 2H), 5.48 (s, 1H), 6.70-7.47 (m, 14H), 10.74 (s, 1H). ^13^C-NMR (CDCl_3_, 100 MHz): δ 16.4, 16.6, 44.4 (d, ^1^*J*_C-P_ = 156.0 Hz), 55.1, (d, ^2^*J*_C-P_ = 6.5 Hz), 62.2 (d, ^2^*J*_C-P_ = 5.5 Hz), 69.3 (d, ^2^*J*_C-P_ = 6.5 Hz), 117.1, 119.4, 124.6, 124.8, 127.9, 128.5, 128.7, 128.9, 129.1, 129.9, 130.2, 130.4, 136.4, 157.2. ^31^P-NMR (CDCl_3_, 161.90 MHz): δ 27.89. HRMS (CI^+^): calculated for C_25_H_30_NO_4_P [M + H]^+^, *m*/*z* 440.1992; found for [M + H]^+^, *m*/*z* 440.2004.

#### 3.4.6. Diethyl({benzyl[(2-hydroxyphenyl)(m-tolyl)methyl]amino}methyl)phosphonate (**3f**)

According to the general procedure, a mixture of 1,3-benzoxazine **1f** (0.3 g, 0.95 mmol), diethyl phosphite (0.13 g, 0.95 mmol, 0.12 mL) and boron trifluoride etherate (0.027 g, 0.19 mmol, 0.024 mL) in acetonitrile (6 mL) was reacted. After purification **3f** (0.15 g, 35%) was obtained as a colorless oil. ^1^H- NMR (CDCl_3_, 400 MHz): δ 1.25 (t, *J* = 7.2 Hz, 3H), 1.32 (t, *J* = 7.2 Hz, 3H), 2.37 (s, 3H), 2.87 (dd, *J* = 15.6, 5.6 Hz, 1H), 2.99 (dd, *J* = 18.0, 15.6 Hz, 1H), 3.66 (d, *J* = 13.2 Hz, 1H), 3.82–4.02 (m, 2H), 4.04–4.17 (m, 2H), 4.09 (d, *J* =13.2 Hz, 1H), 5.41 (s, 1H), 6.70–7.47 (m, 13H), 10.79 (s, 1H). ^13^C-NMR (CDCl3, 100 MHz): δ 16.5 (d, ^3^*J*_C-P_ = 5.0 Hz), 16.6 (d, ^3^*J*_C-P_ = 5.0 Hz), 44.4 (d, ^1^*J*_C-P_ = 155.0 Hz), 55.1, (d, ^3^*J*_C-P_ = 6.0 Hz), 62.3 (d, ^2^*J*_C-P_ = 5.0 Hz), 69.5 (d, ^2^*J*_C-P_ = 6.0 Hz), 117.1, 119.4, 124.7, 127.3, 127.9, 128.6, 128.7, 128.9, 129.1, 129.2, 130.0, 130.3, 131.0, 136.5, 138.5, 157.2. ^31^P-NMR (CDCl_3_, 161.90 MHz): δ 28.52. HRMS (CI^+^): calculated for C_26_H_32_NO_4_P [M + H]^+^, *m*/*z* 454.5404; found for [M + H]^+^, *m*/*z* 454.5454.

#### 3.4.7. Diethyl({benzyl[(2-hydroxyphenyl)(p-tolyl)methyl]amino}methyl)phosphonate (**3g**)

According to the general procedure, a mixture of 1,3-benzoxazine **1g** (0.25 g, 0.79 mmol), diethyl phosphite (0.10 g, 0.79 mmol, 0.10 mL) and boron trifluoride etherate (0.02 g, 0.15 mmol, 0.02 mL) in acetonitrile (5 mL) was reacted. After purification **3g** (0.086 g, 24%) was obtained as a colorless oil. ^1^H-NMR (CDCl_3_, 400 MHz): δ 1.25 (t, *J* = 7.2 Hz, 3H), 1.32 (t, *J* = 7.2 Hz, 3H), 2.37 (s, 3H), 2.84 (dd, *J* = 15.6, 5.2 Hz, 1H), 2.98 (dd, *J* = 18, 16 Hz, 1H), 3.60 (d, *J* = 13.2 Hz, 2H), 3.82–4.00 (m, 2H), 4.044.15 (m, 2H), 5.44 (s, 1H), 6.69–7.36 (m, 13H), 10.83 (s, 1H). ^13^C-NMR (CDCl_3_, 100 MHz): δ 16.5, 16.6, 44.4 (d, ^1^*J*_C-P_ = 157.0 Hz), 55.1, (d, ^3^*J*_C-P_ = 7.0 Hz), 62.3 (d, ^2^*J*_C-P_ = 7.0 Hz), 69.0 (d, ^2^*J*_C-P_ = 7.0 Hz), 117.1, 119.3, 124.7, 127.3, 127.9, 128.7, 129.1, 129.5, 130.0, 130.3, 130.4, 136.5, 138.2, 157.3. ^31^P-NMR (CDCl_3_, 161.90 MHz): δ 22.55. HRMS (CI^+^): calculated for C_26_H_32_NO_4_P [M + H]^+^, *m*/*z* 454.5404; found for [M + H]^+^, *m*/*z* 454.5444.

#### 3.4.8. Diethyl ((benzyl((4-chlorophenyl)(2-hydroxyphenyl)methyl)amino)methyl)phosphonate (**3h**)

According to the general procedure, a mixture of 1,3-benzoxazine **1h** (0.21 g, 0.63 mmol), diethyl phosphite (0.086 g, 0.63 mmol, 0.081 mL) and boron trifluoride etherate (0.017 g, 0.12 mmol, 0.02 mL) in acetonitrile (5 mL) was reacted. After purification **3h** (0.15 g, 50%) was obtained as a colorless oil.

### 3.5. 2-[(3-Cchloropropoxy)(phenyl)methyl]phenol (**5e**)

A mixture of 1,3-benzoxazine **1e** (0.15g, 5 mmol) and 3-chloro-1-propanol (0.5 mL, 0.56g, 5.9 mmol) was stirred at 70 °C for 12 h at 70 °C. Then, the crude was purified by flash chromatography using hexane:EtOAc (80:20), to give **5e** (0.072 g, 53%) ^1^H-NMR (CDCl_3_, 200 MHz): δ 2. (dq, *J* = 6.2, 1.6 Hz, 2H), 3.80 (m, 4H), 5.55 (s, 1H), 7.02 (m, 9H), 7.80 (br, 1H). ^13^C-NMR (CDCl_3_, 50 MHz): δ 32.6, 41.7, 66.4, 85.2, 117.3, 120.0, 124.9, 127.3, 127.6, 128.4, 128.8, 129.0, 129.4, 129.6, 139.9, 155.5. HRMS (CI^+^): calculated for C_16_H_17_O_2_Cl [M + H]^+^, *m*/*z*: 277.0917; found for [M + H]^+^, *m*/*z* 277.0880.

## 4. Conclusions

We have developed a novel “one-pot” method for the synthesis of secondary benzyl phosphonates and α-aminophosphonates from 1,3-benzoxazines. The phosphonates were obtained through direct *o*-QM formation, followed by a phospha-Michael addition reaction and the α-aminophosphonates by iminium ion formation and the subsequent alkylphophites addition. In addition, this synthetic methodology was used to the preparation a valuable *o*-hydroxybenzyl ether derivative, which makes it a useful and efficient method for the synthesis of phosphonates, α-aminophosphonates and benzyl ethers.

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
