# Peer review of "Direct Synthesis of Phosphonates and α-Amino-phosphonates from 1,3-Benzoxazines"

_molecules, 2019, doi:10.3390/molecules24020294_

Round 1

Reviewer 1 Report

The Manuscript (Manuscript ID: molecules-402762) of “Direct Synthesis of Phosphonates and α-Aminophosphonates from 1,3-Benzoxazines” by Oscar Salgado-Escobar and co-workers describes the synthesis of phosphonates and alpha-aminophosphonates using boron trifluoride etherate. Depending on the substrates, the reaction conditions provide two different target products while the major product is aminophosphonates under the reaction conditions. This transformation can be used to construct various secondary and tertiary alpha-aminophosophonates under mild reaction conditions. Despite of the positive tenor, the authors are needed to recheck the data of compound characterization. For example, compound 2c is known compound (Org. Lett. 2017, 19, 5988-5991, compound 3q thereof) but the chemical shift (31P NMR) of 2c shows a large discrepancy with the known data (2c: 27.81 ppm vs 3q: 35.75 ppm). To address this issue, recheck the spectrum and provide a photocopy of the unprocessed original spectrum of compound 2c. In addition, all HRMS data have shown much greater than 5 ppm off. The authors claimed that a catalytic phosphonylation with 20 mol% catalyst loading has been achieved to synthesize compounds 2b, 2c, 2e, 2g, but the product yields are 28%-40% and the NMR spectra show considerable amount of impurities. Therefore, integration values of each peak on all 1H NMR spectra should be provided on each 1H NMR spectrum to evaluate the purity of compounds. It may be premature to claim a catalytic phosphonylation using BF3•OEt2.

Author Response

1.     The authors are needed to recheck the data of compound characterization. For example, compound 2c is known compound (Org. Lett. 2017, 19, 5988-5991, compound 3q thereof) but the chemical shift (31P NMR) of 2c shows a large discrepancy with the known data (2c: 27.81 ppm vs 3q: 35.75 ppm). To address this issue, recheck the spectrum and provide a photocopy of the unprocessed original spectrum of compound 2c.

Response: Attending the suggestions of reviewer, we obtained again the spectra and we included the photocopies of some compounds: RMN 1H, 13C and 31P. However, the chemical shifts are similar to reported in our manuscript previously sent. We obtained the spectrum with internal reference and the other side with external reference, in both cases the results were similar.

 NOTE: In the attached file are the spectra. 

2.     In addition, all HRMS data have shown much greater than 5 ppm off. The authors claimed that a catalytic phosphonylation with 20 mol% catalyst loading has been achieved to synthesize compounds 2b, 2c, 2e, 2g, but the product yields are 28%-40% and the NMR spectra show considerable amount of impurities. Therefore, integration values of each peak on all 1H NMR spectra should be provided on each 1H NMR spectrum to evaluate the purity of compounds.

Response: It has tried to correct the analytical errors. Although the yields on phosphorylation are low, we would like to make them known. A more detailed study is planned with more derivatives using chiral catalysts for enantioselective synthesis.

Reviewer 2 Report

In this paper the authors describe a method for the preparation of phosphonates and a-aminophosphonates from 1,3-benzoxazines through the addition of triethyl phosphite and diethyl phosphite, respectively.

The introduction is well structured and it includes a good number of citations of relevant related articles.

In a previous article (Molecules 2015, 20, 13794-13813) the authors had described a preliminary experiment with a related 1,3-benzoxazine and triethylphosphite without the use of BF3. OEt3.

In a first step, the authors explore the reaction conditions with 1b and triethyl phosphite. Table 1 is missing some relevant experiments in order to understand the real influence on the outcome of the reaction of the type of solvent, the temperature and the catalyst. An experiment using MeCN without BF3.OEt2 at room temperature and one at 82oC would help clarify these issues. Also an experiment with DCM and BF3.OEt2 would help, as in a previously reported experiment, good results had been obtained with this solvent.

In a second step, differently substituted 1,3-benzoxazines are reacted in what is defined as “optimized conditions”, although there is a significant change in the reaction time (72 h) and the equivalents of P(OEt)3 (1 eq) from what is reported in Table 1. The discussion of the effect of the R substituents is not clear. The discussion about steric effects is not convincing as when R= H the a-aminophosphonate is obtained and this is also the case when a bulky alkyl substituent is used (s-Bu). Moreover, in order to draw conclusions from electronic effects, more examples of differently substituted aromatic substituents would be needed.

When reactions are carried out using diethyl phosphite (Scheme 3) , yields are significantly better than when triethylphosphite is used and only a-aminophosphonates are obtained, although no discussion is provided in the text to understand this difference.

The additional reaction to synthesize 5e adds no significant value to the paper. I would remove it from the discussion.

The proposal of the mechanisms is sound but again there is no discussion of the different outcome of the reactions depending on which reagent is used.

Overall, I would suggest that some more experiments are carried out in order to elucidate the effect of the solvent, BF3.OEt2 and reagent used on the reaction mechanism and on its outcome.

The experimental part is well described. I would also suggest to add the NMR spectra for compounds 1c-g.

The use of English is poor and should be checked and corrected. In addition, some other minor errors are listed below:

Reference 15: BA/SR should be Ba/Sr

Scheme 1: The notation under the structure of the products should be 2 and 3, respectively, and not 2b and 3b.

Scheme 2: 24 hours are reported and the experimental part reports 48 h.

Author Response

Reviewer 2

1.      In a previous article (Molecules 2015, 20, 13794-13813) the authors had described a preliminary experiment with a related 1,3-benzoxazine and triethylphosphite without the use of BF3. OEt3. In a first step, the authors explore the reaction conditions with 1b and triethyl phosphite. Table 1 is missing some relevant experiments in order to understand the real influence on the outcome of the reaction of the type of solvent, the temperature and the catalyst. An experiment using MeCN without BF3.OEt2 at room temperature and one at 82oC would help clarify these issues. Also an experiment with DCM and BF3.OEt2 would help, as in a previously reported experiment, good results had been obtained with this solvent.

Response: We agree with the reviewer. In this sense, we studied the effect of the temperature, BF3OEt2 and solvent and the results are explained in the Table 1. We realized 3 experiments: a) an experiment using MeCN without BF3OEt2 at room temperature; b) an experiment using MeCN at 82°C; c) an experiment in DCM and BF3OEt2 (Table 1).

2.      In a second step, differently substituted 1,3-benzoxazines are reacted in what is defined as “optimized conditions”, although there is a significant change in the reaction time (72 h) and the equivalents of P(OEt)3 (1 eq) from what is reported in Table 1.

Response: The change was done. We eliminated the column of time in Table 1 and put the time (72h) in the scheme of Table 1.

3.      The discussion of the effect of the R substituents is not clear. The discussion about steric effects is not convincing as when R= H the a-aminophosphonate is obtained and this is also the case when a bulky alkyl substituent is used (s-Bu). Moreover, in order to draw conclusions from electronic effects, more examples of differently substituted aromatic substituents would be needed.

Response: We agree with the reviewer and we included a discussion of the effect of the R substituents (line 91) considering the stabilization of o-QM and iminium ions substituents.
It is planned to synthesize other enantiomerically pure derivatives using chiral catalysts.

4.      When reactions are carried out using diethyl phosphite (Scheme 3), yields are significantly better than when triethylphosphite is used and only a-aminophosphonates are obtained, although no discussion is provided in the text to understand this difference.

Response: The reviewer is correct: The mechanism of the reaction of the diethylphosphite was included, where the effect of the hydrogen atom of diethylphosphite vs triethylphosphite is observed.

5.      The proposal of the mechanisms is sound but again there is no discussion of the different outcome of the reactions depending on which reagent is used.

Response: In Figure 1 a proposed reaction pathway and discussion for the ring-opening of 1,3-benzoxazines to obtain the phosphonate and a-aminophosphonate was included.

6.      Overall, I would suggest that some more experiments are carried out in order to elucidate the effect of the solvent, BF3.OEt2 and reagent used on the reaction mechanism and on its outcome.

Response: In Table 1 we included others experiments to elucidate the effect of the solvent, BF3OEt2 and reagents used in the ring-opening of 1,3-benzoxazines.

7.      The experimental part is well described. I would also suggest to add the NMR spectra for compounds 1c-g.

Response: The 1c-g spectra of NMR 1H and 13C were added.

8.      Reference 15: BA/SR should be Ba/Sr

Response: The change was done.

9.      Scheme 1: The notation under the structure of the products should be 2 and 3, respectively, and not 2b and 3b.

Response: The change was done.

10.  Scheme 2: 24 hours are reported and the experimental part reports 48 h.

Response: The change was done in Scheme 2.

Reviewer 3 Report

In this manuscript, the authors reported three types of reactions starting from the same substrates 1,3-benzoxazines by change of nucleophiles, triethyl, diethyl phosphite, and alcohol, for the synthesis of phosphonates, α-aminophosphonates and ether. And to explain these results, the authors proposed two possible mechanisms including addition to ortho-Quinone Methide (o-QMs, A) or to iminium ion (B). However, the reactions going through o-QMs (path A) were not discussed clearly.

1)      The intermediates o-QMs and iminium ion were formed under the same conditions, why path A was totally inhibited when using diethyl phosphite as nucleophile compared to triethyl phosphite?

2)      When using triethyl phosphite as nucleophile, why the substituents of the substrates affect the products distribution dramatically, and why these similar substituents (H vs CH3, n-Bu vs s-Bu, Ph vs p-Cl-Ph) gave 100% different products distribution? Line 91, the discussion about bulkier substituent (R= s-Bu) leads to formation the α-aminophosphonate is not proper, because when R=H, smaller group, α-aminophosphonate (89%) was formed. And also, the discussion about electron effect is not very convincing, to prove the electron-withdrawing effect, I think F or NO2 could be better than Cl.

3)      The language needs to be checked carefully, like Line 41, “this compounds”; Line 93-94, “while when the electron-withdrawing group was used afforded the α-aminophosphonate”; Line 120, “could formed”; Line 225, “under reduce pressure”…

4)      The calculated and found HRMS of compounds 2c, 3d, 3c, were not consistent. All the molecular formula in HRMS description need plus H.

5)      Scheme1, “2b” should be “2a-h”, “3b” should be “3a-h”

In short, the manuscript could be reconsidered for publication in Molecules after major revisions.

Author Response

Reviewer 3

1.      The intermediates o-QMs and iminium ion were formed under the same conditions, why path A was totally inhibited when using diethyl phosphite as nucleophile compared to triethyl phosphite?

Response: According the suggestion we have tried to explain the effect.

2.      When using triethyl phosphite as nucleophile, why the substituents of the substrates affect the products distribution dramatically, and why these similar substituents (H vs CH3, n-Bu vs s-Bu, Ph vs p-Cl-Ph) gave 100% different products distribution? Line 91, the discussion about bulkier substituent (R= s-Bu) leads to formation the α-aminophosphonate is not proper, because when R=H, smaller group, α-aminophosphonate (89%) was formed. And also, the discussion about electron effect is not very convincing, to prove the electron-withdrawing effect, I think F or NO2 could be better than Cl.

Response: To explain this we included a discussion of the effect of the R substituents (line 93) considering the stabilization of o-QM and iminium ions substituents and included another experiments in Table 1.

In the beginning we tried to prepare the suggested nitro derivative, however the 2-benzylaminophenol could not be prepared by the methodology used by us. Therefore we could not obtain the corresponding benzoxazine.

3.      The additional reaction to synthesize 5e adds no significant value to the paper. I would remove it from the discussion.

Response: We thank the reviewer suggestion. In this work we describe a direct method to transformation of 1,3-benzoxazines into ethers a versatile precursor to other molecules. With this consideration, we think that this result should be include in the paper as a result of our work.

Round 2

Reviewer 1 Report

December 26, 2018

The author provided sufficient spectral data for the known compounds 3e and 3g and they are matched. However, it is appeared that the accuracy and purity of the compound 3c has not been validated as their 1H NMR spectra, 13C NMR, and 31P NMR do not match with the known compound (Org. Lett. 2017, 19, 5988-5991, compound 3q thereof). Here are the reasonings: a) the 1H NMR spectra show five extra protons in aromatic region (6.5-7.5 ppm) – there must be only five protons in the aromatic region since only one phenyl group is attached on the product, b) there are three extra protons between 3.5-4.2 ppm – there must be only four protons. c) the 13C NMR spectra show extra carbon signals on both aliphatic region and aromatic region, d) the 31P NMR spectra do not match with the known data. The authors are needed to provide further data such as scanned HRMS data to validate their claim if they want to include this compound in this manuscript. Accordingly, the claim of the synthesis of compound 3c should be withdrawn from this manuscript for publication.

Author Response

1.   The author provided sufficient spectral data for the known compounds 3e and 3g and they are matched. However, it is appeared that the accuracy and purity of the compound 3c has not been validated as their 1H NMR spectra, 13C NMR, and 31P NMR do not match with the known compound (Org. Lett. 2017, 19, 5988-5991, compound 3q thereof). Here are the reasonings: a) the 1H NMR spectra show five extra protons in aromatic region (6.5-7.5 ppm) – there must be only five protons in the aromatic region since only one phenyl group is attached on the product, b) there are three extra protons between 3.5-4.2 ppm – there must be only four protons. c) the 13C NMR spectra show extra carbon signals on both aliphatic region and aromatic region, d) the 31P NMR spectra do not match with the known data. The authors are needed to provide further data such as scanned HRMS data to validate their claim if they want to include this compound in this manuscript. Accordingly, the claim of the synthesis of compound 3c should be withdrawn from this manuscript for publication.

Response: Attending yours suggestions, the compound was withdrawn and the language was corrected.

Reviewer 2 Report

The authors have addressed most of the comments made in the first peer-review.

I would still recommend extensive language revision.

Author Response

1.   The authors have addressed most of the comments made in the first peer-review. 

I would still recommend extensive language revision.

Response: We appreciate yours comments, the language corrections were made.

Reviewer 3 Report

In the revised manuscitpt, the authors addressed the reviewer's concerns. So I think it can be considered for publication in Molecules in present form. 

Author Response

1.   In the revised manuscript, the authors addressed the reviewer's concerns. So I think it can be considered for publication in Molecules in present form.

    Response: We appreciate yours recommendations, the language revision was carried out.